

# Development of a homogeneous time-resolved FRET (HTRF) assay for the quantification of Shiga toxin 2 produced by *E. coli*

Keiji Nakamura[1], Chikashi Tokuda[2], Hideyuki Arimitsu[3], Yoshiki Etoh[4], Mitsuhiro Hamasaki[4], Yuichiro Deguchi[5], Itsuki Taniguchi[1], Yasuhiro Gotoh[1], Yoshitoshi Ogura[6] and Tetsuya Hayashi[1]

[1] Department of Bacteriology, Graduate School of Medical Sciences, Kyushu University, Fukuoka, Japan
[2] Cisbio K.K., Chiba, Japan
[3] School of Human Science and Environment, University of Hyogo, Himeji, Japan
[4] Fukuoka Institute of Health and Environmental Sciences, Dazaifu, Japan
[5] Production Medicine Center, Agricultural Mutual Aid Association in Miyazaki Prefecture, Koyugun-Shintomicho, Japan
[6] Division of Microbiology, Department of Infectious Medicine, Kurume University School of Medicine, Kurume, Japan

Corresponding author
Tetsuya Hayashi,
thayash@bact.med.kyushu-u.ac.jp

## ABSTRACT

Shiga toxin-producing *Escherichia coli* (STEC) is a major intestinal pathogen and causes serious gastrointestinal illness, which includes diarrhea, hemorrhagic colitis, and life-threatening hemolytic uremic syndrome. The major virulence factors of STEC are Shiga toxins (Stx1 and Stx2), which belong to the AB-type toxin family. Among several subtypes of Stx1 and Stx2, the production of Stx2a is thought to be a risk factor for severe STEC infections, but Stx2a production levels vary markedly between STEC strains, even strains with the same serotype. Therefore, quantitative analyses of Stx2 production by STEC strains are important to understand the virulence potential of specific lineages or sublineages. In this study, we developed a novel Stx2 quantification method by utilizing homogeneous time-resolved fluorescence resonance energy transfer (HTRF) technology. To determine suitable "sandwich" assay conditions, we tested 6 combinations of fluorescence-labeled monoclonal antibodies (mAbs) specific to Stx2 and compared the HTRF signal intensities obtained at various incubation times. Through this analysis, we selected the most suitable mAb pair, one recognizing the A subunit and the other recognizing the B subunit, thus together detecting Stx holotoxins. The optimal incubation time was also determined (18 h). Then, we optimized the concentrations of the two mAbs based on the range for linearity. The established HTRF assay detected 0.5 ng/ml of the highly purified recombinant Stx2a and Stx2e proteins and the working range was 1–64 ng/ml for both Stx2a and Stx2e. Through the quantification analysis of Stx proteins in STEC cell lysates, we confirmed that other Stx2 subtypes (Stx2b, Stx2c, Stx2d and Stx2g) can also be quantified at a certain level of accuracy, while this assay system does not detect Stx2f, which is highly divergent in sequence from other Stx2 subtypes, and Stx1. As the HTRF protocol we established is

simple, this assay system should prove useful for the quantitative analysis of Stx2 production levels of a large number of STEC strains.

# INTRODUCTION

Shiga toxin-producing *Escherichia coli* (STEC) causes diarrhea and hemorrhagic colitis with life-threatening complications, such as hemolytic uremic syndrome (HUS). Many serogroups of STEC, such as O157, O26, O111, O103, O121 and O145, cause outbreaks and sporadic cases worldwide (*Centers for Disease Control & Prevention (CDC), 2018*; *European Food Safety Authority & European Centre for Disease Prevention & Control (EFSA & ECDC), 2019*; *National Institute of Infectious Diseases (NIID), 2018*; *Smith et al., 2019*). While STEC possesses a broad range of virulence factors, Shiga toxin (Stx, previously called Verotoxin) is the key virulence factor belonging to the AB-type toxin family. Stxs are classified into two immunologically distinct types, Stx1 and Stx2, and both types include several subtypes (Stx1a, Stx1c and Stx1d; Stx2a, Stx2b, Stx2c, Stx2d, Stx2e, Stx2f and Stx2g) (*Scheutz et al., 2012*). Among these subtypes, Stx1 subtypes exhibit high amino acid sequence similarity (>95% identity). In contrast, Stx2 subtypes exhibit a notable degree of intragroup diversity. In particular, Stx2f is highly divergent from other Stx2 subtypes (70.9–77.0% amino acid sequence identities; Fig. S1).

Several biological and epidemiological studies have revealed that Stx2a production is associated with more severe diseases and a higher risk of HUS (*Bielaszewska et al., 2013*; *Dallman et al., 2015*; *Fuller et al., 2011*). In addition, it has been suggested that the Stx2a production level may be linked to the high virulence potential of a specific lineage of STEC O157 (*Ogura et al., 2015*). It has also been reported that Stx2a production levels vary markedly between STEC strains, even between strains with the same serotype (*i.e.* STEC O157, O26 and O145) (*Carter et al., 2021*; *Ishijima et al., 2017*; *Ogura et al., 2018*). Therefore, quantitative analyses of the Stx2 production level are important to understand the virulence potentials of specific lineages or sublineages. Several immunological or molecular genetic methods for the detection of Stx proteins or *stx* genes in STEC isolates have been developed (*Parsons et al., 2016*). However, there are only a few reports that described methods to quantify Stx (*Yamasaki et al., 2015*; *Quiñones et al., 2009*). Among these methods, sandwich ELISA-based methods exhibit high sensitivity and a broad dynamic range (*He et al., 2018*; *Yamazaki et al., 2013*) but require multiple steps to obtain final data; thus, they are sometimes difficult to use for the analysis of a large number of samples. There are also potential risks of technical errors due to complicated protocols. Now in Japan, the ELISA kits for Stx quantification, such as RIDASCREEN Verotoxin (R-Biopharm AG, Darmstadt, Germany), are not commercially available, and thus a kit based on the reversed passive latex agglutination (RPLA)-based

assay (VTEC-RPLA) is used for the quantitative analysis of Stx. As the protocol of this assay is simpler than ELISA, this assay has been widely used (*Beutin, Zimmermann & Gleier, 1996*; *Karmali, Petric & Bielaszewska, 1999*; *Chart, Willshaw & Cheasty, 2001*; *Ogura et al., 2015*; *Rahman et al., 2018*). However, it is a 2-fold dilution endpoint assay (thus semiquantitative assay) and endpoints are determined based on the occurrence of agglutination subjectively by users. Therefore, it is sometimes difficult to precisely determine toxin concentrations with high reproducibility between laboratories and even in a laboratory. In this study, we developed a novel quantification method of Stx2 by utilizing homogeneous time-resolved fluorescence resonance energy transfer (HTRF) technology (*Degorce et al., 2009*) to quantitatively analyze the Stx2 production levels of isolated STEC strains. This technology was already applied to the detection of other bacterial toxins, such as the protective antigen of *Bacillus anthracis* (*Cohen et al., 2014*). A fluorescence resonance energy transfer (FRET)-based Stx2 detection system (named AlphaLISA assay) has also been developed (*Armstrong et al., 2018*). The method we developed is simple, requiring only mixing samples with labeled antibodies and incubating them for a certain time, and the high reproducibility with a wide calibration range of the method was also confirmed.

## METHODS

### Stx2 toxins and antibodies

We used two types of Stx2a preparations as standards. One was a commercially available lyophilized Stx2a preparation (Verotoxin 2; Denka Seiken, Tokyo, Japan) that was used for optimizing assay conditions. The other was a highly purified recombinant Stx2a preparation, which we used to evaluate the established assay. This recombinant Stx2a was expressed as a holotoxin and purified as previously described (*Arimitsu, Sasaki & Tsuji, 2016*). In addition, we used a commercially available Stx1 preparation (Verotoxin-1; Nacalai Tesque, Kyoto, Japan), a highly purified recombinant Stx2e protein that was prepared as previously described (*Arimitsu et al., 2013a*), and a commercially available recombinant Stx2f preparation (Abraxis, CA, USA). The recombinant Stx2f protein contains an amino acid substitution at the active site in the A subunit (glutamic acid at 167) and is thus enzymatically inactive.

For developing an HTRF assay, we used three monoclonal antibodies (mAbs): MBS313194 (anti-Stx1/Stx2 B-subunit, MyBioSource), LS-C132889 (anti-Stx2 B-subunit; Lifespan Biosciences, Seattle, WA, USA), and 20273-04 (anti-Stx2 A-subunit; Nacalai Tesque, Kyoto, Japan). These mAbs were fluorescently labeled using the Europium Cryptate labeling kit or the d2 labeling kit (both from Cisbio/PerkinElmer, Waltham, MA, USA). Donor and acceptor antibodies were labeled with europium (Eu) cryptate and d2, respectively.

### Bacterial strains and preparation of cell lysates

The STEC strains used in this study are listed in Table 1. They were isolated from humans or animals in Japan and contained the *stx2* gene alone or along with the *stx1* gene. Except for the five strains newly sequenced in this study (strains 07E033, 10E094, 11E007,

**Table 1** STEC strains used in this study and their Stx2 production levels determined by HTRF and RPLA assay.

| Strain | Serotype | Source | *stx* subtype | Stx2 titer | | Accession No. | References |
|---|---|---|---|---|---|---|---|
| | | | | HTRF (ng/ml)* | RPLA (unit/ml)† | | |
| Sakai | O157:H7 | Human | *stx1a/stx2a* | 9,900 ± 520 | 8,092 | BA000007 | *Hayashi et al. (2001)*. |
| 130549 | O26:H11 | Human | *stx2a* | 34,000 ± 1,500 | 32,768 | BDHP01000185 | *Ishijima et al. (2017)*. |
| 11E007 | O146:H21 | Human | *stx2b* | 330 ± 39 | 256 | BOUS01000033 | This study |
| SI-NP059 | Onovel1:H16 | Bovine | *stx2b* | 1,500 ± 120 | 512 | BGIA01000167 | *Arimizu et al. (2019)*. |
| WGPS6 | O157:H7 | Human | *stx2c* | 2,700 ± 79 | 2,048 | AP012539 | *Ogura et al. (2015)*. |
| 5044 | O145:H28 | Human | *stx2c* | 2,800 ± 160 | 2,048 | BJPC01000106 | *Nakamura et al. (2020)*. |
| 112808 | O145:H28 | Human | *stx1a/stx2d* | 1,200 ± 400 | 1,024 | BJQB01000217 | *Nakamura et al. (2020)*. |
| 10E094 | OUT:H7 | Human | *stx2d* | 85 ± 5 | 32 | BOUT01000015 | This study |
| EC-PM083 | O157:H19 | Swine | *stx2e* | 640 ± 90 | 512 | BOUQ01000003 | This study |
| EC-PM098 | O50:H32 or O2:H32 | Swine | *stx2e* | 100 ± 20 | 64 | BOUR01000009 | This study |
| 07E033 | O63:H6 | Human | *stx2f* | N.D. | 128 | BOUS01000033 | This study |
| KS-P022 | O168:H8 | Bovine | *stx2g* | 890 ± 170 | 512 | BGAG01000027 | *Arimizu et al. (2019)*. |
| MG1655 | O16:H48 | Widely used laboratory strain | *stx*-negative | N.D. | N.D. | U00096 | *Blattner et al. (1997)*. |

**Notes:**
* Technical triplicate (mean value ± SD).
† 1 unit/ml roughly corresponds to 1 ng/ml of Stx2a (see "Methods")
UT, untypable.
N.D., not detected.

EC-PM083 and EC-PM098), the *stx* subtypes in all STEC strains were determined previously (references are available in Table 1). We included an *stx*-negative *E. coli* strain (MG1655; accession no. U00096) in the strain set. Bacterial cells were cultured at 37 °C with shaking in lysogeny broth (LB) medium. To prepare cell lysates, STEC overnight cultures were inoculated in 2 mL of LB broth at a cell concentration of 0.1 $OD_{600}$ and grown to mid-log phase. Then, mitomycin C (MMC; Kyowa Kirin, Tokyo, Japan) was added to the cultures (0.5 μg/ml at the final concentration). After a 6-h incubation with MMC, cultures were subjected to four 75-s periods of sonication with a Bioruptor (Cosmo Bio, Tokyo, Japan) to disrupt the STEC cells in cultures. The sonicated samples were centrifuged at 7,700×*g* for 10 min, and the supernatants were used as Stx2-containing cell lysates. Cell lysates without MMC-treatment were also prepared similarly but without the addition of MMC.

## Optimization of the HTRF assay system

For optimizing assay conditions (the combination of mAbs, incubation time, and concentrations of fluorescence-labeled mAbs), the aforementioned lyophilized Verotoxin 2 and three mAbs were used. The toxin in one vial was dissolved in 0.25 ml of phosphate buffered saline containing 0.2% BSA and a preservative (0.1% $NaN_3$ or 0.05% ProClin300) and serially diluted with the same buffer. Diluted toxin preparation (10 μl per reaction) was mixed with serially diluted Eu- and d2-labeled mAbs (5 μl each per reaction; diluted with HTRF detection buffer (Cisbio, Waltham, MA, USA)) in all possible

combinations, and the mixtures were incubated at room temperature. After a 0.5–18 h incubation, the emissions at 665 nm and 620 nm were measured by a PHERAstar FS microplate reader (BMG LABTECH, Ortenberg, Germany). Delta F values (%) of each reaction were calculated for interassay comparison following the procedure described on the CisBio website (https://www.cisbio.jp/content/signal-treatment-and-analysis/).

## Determination of Stx2 concentrations by the optimized HTRF assay

Diluted recombinant Stx2a preparation (used as a standard) or cell lysates were mixed with the d2-labeled anti-Stx2 A-subunit mAb (20273-04, 0.8 μg/ml at the final concentration) and the Eu-labeled anti-Stx2 B-subunit mAb (LS-C132889, 0.5 μg/ml at the final concentration). After incubation for 18–24 h at room temperature, emissions were measured at 665 nm and 620 nm by an Infinite 200 PRO (TECAN, Männedorf, Switzerland), and the ratio and delta ratio (DR) were calculated as follows:

$$\text{Ratio} = (665 \text{ nm counts}/620 \text{ nm counts}) \times 100$$

$$\text{DR} = \text{average Ratio}_{(\text{Sample})} - \text{average Ratio}_{(\text{Diluent background})}$$

Average ratios were obtained by two measurements using independently prepared Stx2 samples. Based on the standard values prepared using Stx2a, curve fitting for each assay was performed with a weighted four parameter logistic (4PL) model using Prism 8 software (GraphPad Software, San Diego, CA, USA), and Stx2 concentrations in cell lysates were calculated with this curve-fit model using the same software.

## RPLA assay

Stx2 concentrations in each STEC cell lysate were also semiquantitatively evaluated by the RPLA assay using the VTEC-RPLA kit (Denka Seiken, Tokyo, Japan) according to the manufacturer's instructions. This kit detects a wide range of Stx2 subtypes (including Stx2f) by capturing Stx2 proteins by latex beads coated with rabbit anti-Stx2 polyclonal antibody (pAb), and relative concentrations are expressed as titers (the highest dilution that generates agglutination). In this study, Stx2 concentrations measured by this kit were expressed as units/ml (one unit/ml is the lowest concentration that generates agglutination). The detection limit of the kit was approximately 1 ng/ml when Stx2a was used; thus, one unit/ml corresponded to 1 ng/ml.

## Genome sequencing of STEC strains

Genomic DNA of five STEC strains, 11E007 (*stx2b*-positive), 10E094 (*stx2d*-positive), 07E033 (*stx2f*-positive), EC-PM083 and EC-PM098 (*stx2e*-positive), was purified using a DNeasy Blood and Tissue Kit (Qiagen, Hilden, Germany). DNA libraries of the three STEC strains (*stx2b*-, *stx2d*-, and *stx2f*-positive strains) and the two *stx2e*-positive strains were prepared using the QIAseq FX DNA Library Kit (Qiagen, Hilden, Germany) and NEBNext Ultra II FS DNA Library Prep Kit for Illumina (New England Biolabs, Ipswich, MA, USA), respectively. Illumina sequencing and sequence assembly were performed as
previously described (*Nakamura et al., 2020*). The serotypes of all sequenced strains were determined by SerotypeFinder 2.0 (https://cge.cbs.dtu.dk/services/SerotypeFinder/) (*Joensen et al., 2015*). Subtyping of *stx* genes was performed by BLASTN search (>98.5% identity with >99% coverage) with the reference sequences described previously (*Scheutz et al., 2012*). The draft genome sequences of these five strains have been deposited in DDBJ/EMBL/GenBank under BioProject accession numbers starting from PRJDB8147. All accession numbers of *stx2*-encoding sequences in these strains are listed in Table 1.

## RESULTS AND DISCUSSION

### Establishment of an HTRF assay system for quantification of Stx2

To detect Stx2, we employed a "sandwich" assay in which HTRF signals are generated through energy transfer from donor to acceptor that are labeled on the antibodies. Three commercially available mAbs were used as donors or acceptors in this study, and donors and acceptors were labeled with Eu cryptate and d2, respectively. We first compared the HTRF signal intensities in all combinations of mAbs (6 combinations) at various incubation times (0.5–18 h) to determine the best pair for the sandwich assay (Fig. 1A). Three pairs (Donor/Acceptor: 20273-04/LS-C132889, 20273-04/MBS313194 and MBS313194/LS-C132889) generated no or weak signals. Among the remaining three pairs, the signals of the LS-C132889/20273-04 pair were the highest at 18 h at each concentration of Stx2. In addition, this pair detected only holotoxin because LS-C132889 recognizes the A subunit and 20273-04 recognizes the B subunit. Therefore, we concluded that 18 h of incubation with Eu-labeled LS-C132889 and d2-labeled 20273-04 mAbs was the most suitable assay condition. It should be noted that although we employed 18-h incubation in this study as the strongest signals were obtained, it seems possible to shorten incubation time if necessary although the working range may be changed.

We next optimized the antibody concentrations. Under a fixed concentration of d2-labeled mAb (1.0 µg/ml), the range for linearity in HTRF assays was evaluated using twofold serially diluted Eu-labeled mAb by visual inspection of a plot of signals (Fig. 1B). The widest linear range was observed when 0.5 µg/ml Eu-labeled mAb was used. Therefore, using this concentration of Eu-labeled mAb, we performed a similar analysis to determine the optimal concentration of d2-labeled mAb. Based on the results of this analysis, we decided to use 0.8 µg/ml d2-labeled mAb for the assay.

### Evaluation of the dynamic range and specificity of the HTRF assay using purified Stx preparations (Stx2a, Stx2e, Stx2f, and Stx1a)

Next, we evaluated the dynamic range and specificity of the established HTRF assay using four purified Stx preparations. Note that among the four preparations, the Stx2a and Stx2e preparations were highly purified to homogeneity (*Arimitsu et al., 2013a*, *2013b*).

**Stx2a:** The HTRF assay detected 0.5 ng/ml Stx2a (Table S1), and the signal intensity reached a plateau at 128 ng/ml (Fig. 2). Although a calibration curve could be drawn using the signals ranging between the lower and upper limits (0.5–128 ng/ml) (Fig. S2), signals around the lower limit were very weak (DR: 0.6–0.8), and the deviation of signals at 128 ng/ml was relatively larger than those at other concentrations (DR: 76.1–91.5;

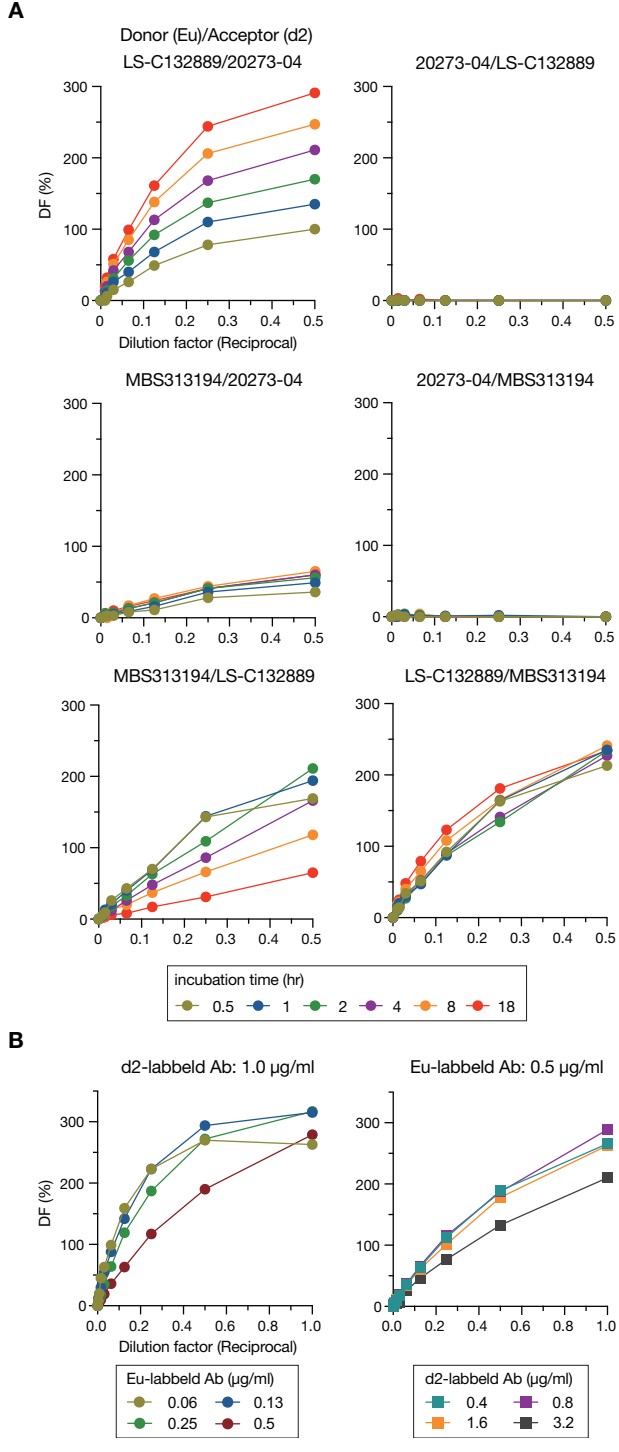

**Figure 1 Optimization of the HTRF assay for Stx2 detection.** The HTRF assay conditions were optimized by selecting the most suitable antibody pair and incubation time (A) and determining the most appropriate concentrations of antibodies (B). The relative amount of Stx toxin (commercially available lyophilized Stx2; see "Methods") used in each assay is presented as the reciprocal of the final dilution factor on the X-axis. Delta F (DF) values were used as the signal intensities of each reaction. All data are presented as the mean values (n = 3).

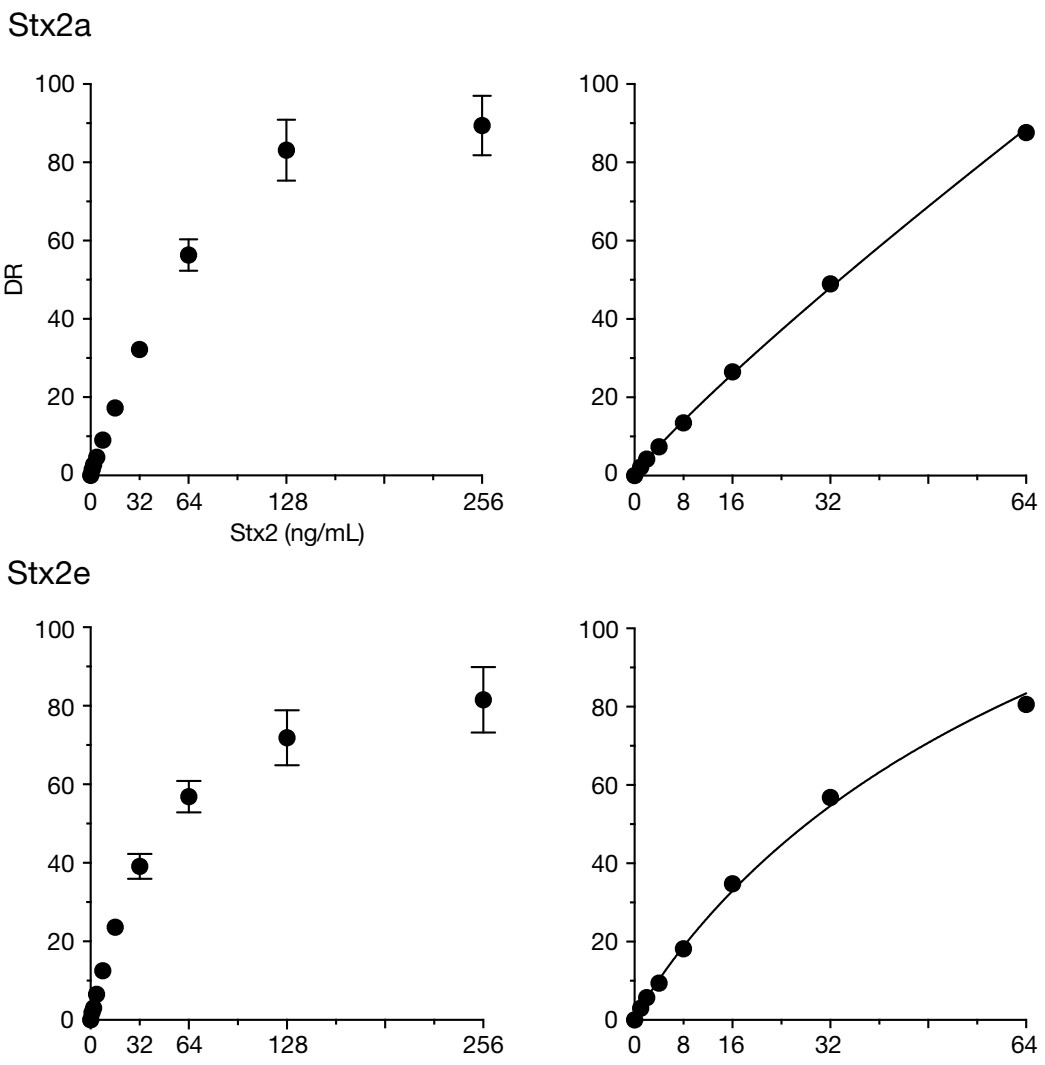

**Figure 2 Dynamic ranges of the HTRF assay for purified Stx2a and Stx2e.** In the left panels, delta ratio (DR) values at each Stx2 concentration ranging from 1 ng/ml to 256 ng/ml are shown. All measurements were conducted in triplicate, and data are presented as the mean values ± standard deviation. In the right panels, DR values at Stx2 concentrations ranging from 1 ng/ml to 64 ng/ml in a representative assay are shown. A weighted four parameter logistic (4PL) curve can be applied to the analysis of the DRs in this range of Stx2 concentrations.

SD = 7.8) (Table S1). By narrowing the range of Stx2 (1–64 ng/ml), more reproducible signal intensity and improved linearity were achieved (Table S1 and Fig. 2). Therefore, the working range was determined to be between 1 and 64 ng/ml.

**Stx2e:** In our HTRF assay, the detection limit of Stx2e was identical to that of Stx2a, and the working range was also similar to that of Stx2a (Fig 2, Fig. S2 and Table S2). Thus, our HTRF assay can quantify Stx2e with a dynamic range and sensitivity similar to those of Stx2a.

**Stx2f and Stx1a:** We further examined the applicability of the HTRF assay for Stx2f and Stx1a, the former of which is highly divergent in amino acid sequence from other Stx2 subtypes (Fig. S1). Stx2f signals were not detected even at a high concentration

(1.28 µg/ml) (Table S2). Furthermore, Stx2f in the cell lysate of strain 07E033, a Stx2f-producing strain, was not detected by the HTRF assay, although the presence of Stx2f in the lysate was confirmed by the VTEC-RPLA assay (Table 1). This result is consistent with a previous report that the antigenicity of the A subunit of Stx2f is different from that of other Stx2 subtypes (*Skinner et al., 2013*) and suggests that the mAb against the A subunit (LS-C132889) used in our HTRF assay is unable to capture Stx2f. Although patients infected with Stx2f-producing strains usually show milder symptoms (*Friesema et al., 2014*), there are a few reports of the isolation of Stx2f-producing *E. coli* from HUS patients (*Friesema et al., 2015*; *Grande et al., 2016*), and thus the virulence potential of Stx2f-producing strains has not yet been well elucidated. As expected, Stx1a was not detected by the HTRF assay (Table S2).

## Applicability of the HTRF assay to other Stx2 subtypes

Finally, we examined whether the HTRF assay can be applied to Stx2b, Stx2c, Stx2d and Stx2g, as it was to Stx2a and Stx2e. As purified preparations were not available for these four subtypes, we examined them using MMC-treated STEC cell lysates, each containing Stx2b, Stx2c, Stx2d or Stx2g. For comparison, Stx2a-, Stx2e- and Stx2f-containing cell lysates and that of an *stx*-negative strain (MG1655) were also included in this analysis. Note that before this analysis, we confirmed that each of the STEC strains used possessed a single *stx2* gene by BLASTN search of whole genome sequences, five of which were determined in this study.

As shown in Table 1, the HTRF assay detected the four Stx2 subtypes in each STEC lysate and provided their estimated concentrations in each lysate. No signal intensity was obtained in the cell lysate of the *stx*-negative *E. coli* strain. We then measured the concentrations of the Stx2 subtypes using the VTEC-RPLA system with Stx2a as a standard (see "Methods") (Table 1) and compared them with those estimated by the HTRF assay. The results indicated that the Stx concentrations estimated by the two systems correlated very well (Pearson's $r = 0.9989$, $P < 0.0001$; Fig. 3).

In the comparison of amino acid sequences of mature Stx proteins (leader signal-removed) with those of Stx2a, there were 3-92 amino acid substitutions in Stx2 subtypes (Fig. S1): 19 (14 in A-subunit + 5 in B-subunit) in Stx2b, 3 (1 + 2) in Stx2c, 4 or 5 (2 or 3 + 2) in Stx2d, 26 (17 + 9) in Stx2e, 92 (83 + 9) in Stx2f, and 18 (14 + 4) in Stx2g. Thus, except for Stx2f, Stx2e is most dissimilar to Stx2a in amino acid sequence (92.9% identity). As already described, the HTRF signals of Stx2e were nearly identical to those of Stx2a (Fig. 2). These results suggest that the two mAbs used in this system recognize the epitopes common to Stx2a and Stx2e. The numbers of amino acid residues in the four Stx2 subtypes (Stx2b, Stx2c, Stx2d and Stx2g), different from those shared by Stx2a and Stx2e, were less than 10 residues in the A-subunits of each subtype (10 in Stx2b, 1 in Stx2c, 0 or 1 in Stx2d, and 7 in Stx2g) and one residue in the B-subunits of each subtype. Such a high similarity in primary structure between Stx2 subtypes except for Stx2f suggests that they exhibit very similar reactivities to the two mAbs. Thus, the concentrations of Stx2 subtypes estimated by the HTRF assay using Stx2a as a standard most likely represent concentrations very close to the true concentrations, which could be determined by using

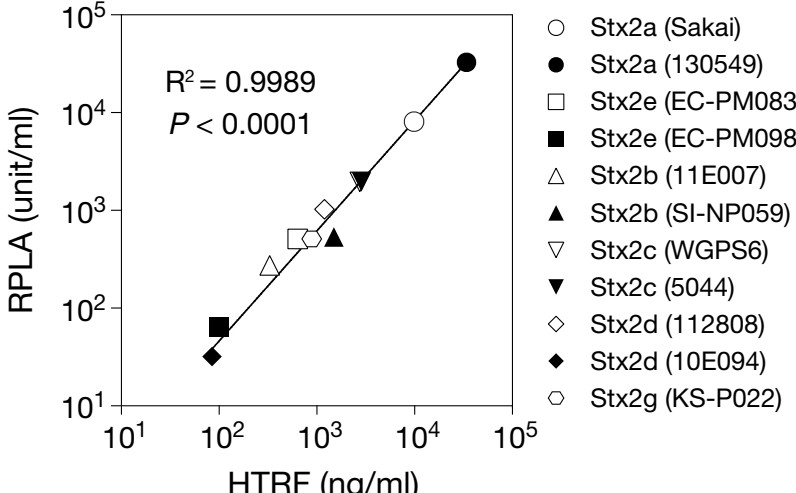

**Figure 3 Relationship between the Stx2 concentrations estimated by the HTRF and RPLA assays.**
The concentration of Stx2 produced by each STEC strain (shown in parentheses) was measured by HTRF and RPLA assays. Each plot represents the mean values of Stx2 concentrations determined by three independent HTRF assays ($X$-axis) and the RPLA unit obtained from a single assay ($Y$-axis). Each value is shown in Table 1. See Methods for the definition of units in the RPLA assay. Note that rabbit anti-Stx2 pAb was used in the RPLA assay. Both axes are displayed in logarithmic scale. The solid line represents the nonlinear regression. The correlation between the data from the two assay systems was evaluated using the Pearson correlation coefficient, as shown in the graph.

the cognate Stx subtype as a reference. The high-level correlation of the Stx concentrations estimated by the HTRF assay with those estimated by the pAb-based VTEC-RPLA system shown in Fig. 3 gives some support to this assumption, although small structural differences between the Stx2 subtypes could affect the signal intensities to some extent in the assay due to the differences in affinities of antibodies to antigens and/or physical distances between donors and acceptors.

Finally, we measured the concentrations of Stx2a, Stx2e and Stx2g in the cell lysates which were prepared without MMC treatment from strains Sakai, EC-PM083 and KS-P022, respectively, using the HTRF assay we developed. While the concentrations were much lower than those with MMC-treatment as expected, the concentrations of these toxins were successfully determined (68.0 ± 3.9 ng/ml in Sakai, 20.6 ± 0.3 ng/ml in EC-PM083, and 5.0 ± 0.8 ng/ml in KS-P022; data from three independent experiments presenting the mean value ± SD).

## CONCLUSION

We developed the HTRF assay for quantifying several Stx2 subtypes. While this assay system does not detect Stx2f, Stx2e can be quantified similarly to Stx2a, and other Stx2 subtypes (Stx2b, 2c, 2d and 2g) can also be quantified at a certain level of accuracy. The HTRF protocol is simple, avoiding technical errors. In addition, this assay is not an endpoint assay like RPLA, and its working range (1–64 ng/ml for Stx2a and Stx2e) is likely wide enough. Due to these advantages, this assay system is applicable to the analysis of Stx2 production levels of a large number of STEC strains. It should be noted, however,

that although this assay may be applicable to the quantification of these toxins in clinical and environmental samples, careful validations are required before applying this assay system to these samples. Another limitation of this study is a lack of comparison of this system with other quantitative assay systems to evaluate it more precisely due to the restriction on their importation to our country.

## ACKNOWLEDGEMENTS

We thank M. Horiguchi and K. Ozaki for providing technical assistance. We also thank N. Ishijima and R. Tohya for providing the Stx2f toxoid and *stx2e*-positive STEC strains, respectively.

### Funding

This research was supported by AMED under Grant Number 20fk0108065h0803 and 21fk0108611h0501 to Tetsuya Hayashi, and a KAKENHI from the Japan Society for the Promotion of Science (No. 18K07116) to Keiji Nakamura. The funders had no role in study design, data collection and analysis, decision to publish, or preparation of the manuscript.

### Grant Disclosures

The following grant information was disclosed by the authors:
AMED: 20fk0108065h0803 and 21fk0108611h0501.
KAKENHI: 18K07116.

### Competing Interests

Chikashi Tokuda is employed by Cisbio K.K., we asked Cisbio for their assistance in developing the HTRF assay system and his participation in our work as a collaborator.

### Author Contributions

- Keiji Nakamura conceived and designed the experiments, performed the experiments, analyzed the data, prepared figures and/or tables, authored or reviewed drafts of the paper, and approved the final draft.
- Chikashi Tokuda performed the experiments, authored or reviewed drafts of the paper, and approved the final draft.
- Hideyuki Arimitsu performed the experiments, authored or reviewed drafts of the paper, and approved the final draft.
- Yoshiki Etoh performed the experiments, authored or reviewed drafts of the paper, and approved the final draft.
- Mitsuhiro Hamasaki performed the experiments, authored or reviewed drafts of the paper, and approved the final draft.
- Yuichiro Deguchi performed the experiments, authored or reviewed drafts of the paper, and approved the final draft.

- Itsuki Taniguchi analyzed the data, authored or reviewed drafts of the paper, and approved the final draft.
- Yasuhiro Gotoh analyzed the data, authored or reviewed drafts of the paper, and approved the final draft.
- Yoshitoshi Ogura analyzed the data, authored or reviewed drafts of the paper, and approved the final draft.
- Tetsuya Hayashi conceived and designed the experiments, prepared figures and/or tables, authored or reviewed drafts of the paper, and approved the final draft.

## DNA Deposition

The following information was supplied regarding the deposition of DNA sequences:

All accession numbers of *stx2*-encoding sequences are available at GenBank: BA000007, BDHP01000185, BOUS01000033, BGIA01000167, AP012539, BJPC01000106, BJQB01000217, BOUT01000015, BOUQ01000003, BOUR01000009, BOUS01000033, and BGAG01000027 (see Table 1).

## Data Availability

The draft genome sequences of the five strains sequenced in this study (07E033, 10E094, 11E007, EC-PM083 and EC-PM098) are available from DDBJ/EMBL/GenBank: PRJDB8147.

## Supplemental Information

Supplemental information for this article can be found online at http://dx.doi.org/10.7717/peerj.11871#supplemental-information.

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
