# Peer review of "Development of a homogeneous time-resolved FRET (HTRF) assay for the quantification of Shiga toxin 2 produced by E. coli"

_PeerJ, doi:10.7717/peerj.11871_

## Round 0.1 · original submission · Major Revisions

Please take particular attention to the Reviewer's 2 comments, which I think are relevant and must be experimentally addressed.

Reviewer 1 ·

Basic reporting

This report by Nakamura et al, describes the development of a novel quantification method to quantify Shiga toxin 2, an important virulence factor of STEC strains associated with more severe diseases. This new methodology will be a valuable tool to better understand the virulence potential of specific strains in order to control STEC life-threatening complications.

Please find below some comments I suggest to help the readers understanding the general background and objective of this study.

1. I suggest to present a brief description in INTRODUCTION of the using of the HTRF assay for the quantification of toxins: explain that this is not the first report to apply this technology in bacterial toxin detection. Please give some examples, i.e., Cohen et al., DOI 10.1007/s10895-014-1354-7, for the detection of Bacillus anthracis PA.

2. Present also in the Introduction Section, a background of the methods available for Stx2 quantification including the VTEC-RPLA kit presented in results. Which is the disadvantage of the VTEC-RPLA kit? VTEC-RPLA seems to be also a simple method that does not require several steps to obtain the final data, thus, RPLA is applicable to a large number of samples. Please, explain why there is a need to develop a new method.

3. From the clinical and practical point of view, it will be important to investigate if the methodology present in this study can detect and quantify Stx2 in stool samples and/or in STEC culture supernatants (without any additional treatment, such as mitomycin C treatment and sonication). Have the authors consider this possibility?

4. In the paragraph of “Applicability of the HTRF assay to other Stx2 subtypes”, discuss the consequences of overlooking Stx2f by HTRF. Are Stx2f-producing STEC strains associated with less severe disease?

5. Conclusion, line 250: .... HTRF assay for quantifying “Stx2a”.... Should be “Stx2” or “several Stx2 subtypes”.
(In Table 1, the production level of all Stx2 subtypes, except Stx2f, are presented).

6. Conclusion, line 253. Please explain the disadvantage of “an endpoint assay to detect Stx2.”

Experimental design

No comment

Validity of the findings

No comment

·

Basic reporting

The paper by Nakamura et al. describes a novel assay that relies on homogeneous time-resolved fluorescence resonance energy transfer (HTRF) for the quantitation of Shiga toxin 2 produced by E. coli.

The paper is well written and the methods employed are sufficiently clear; the results are also properly described and the figures are of an appropriate quality. The raw data is also accessible as per journal standards.

I noted a couple of typing/grammatical errors:

-Line 27: "The major virulence factor of STEC is Shiga toxin (Stx1 and Stx2)" should be "The major virulence factors of STEC are Shiga toxins (Stx1 and Stx2)"

-Line 120: There is an additional parenthesis.

Experimental design

I have the following concerns about the experimental design:

1-The authors should have compared their quantitative data output from the HTRF technique to that from another quantitative assay, such as the widely available ELISA for these toxins. Comparisons to a semi-quantitative assay, with an output in 'units', such as reverse passive latex agglutination (RPLA) does not allow for proper evaluation.

2-There is another fluorescence resonance energy transfer-based assay for assessing levels of Shiga toxins, albeit with a narrower range (based on the literature), as indicated by the authors. How does this assay compare in a lab setting, in practice, to the one developed by the authors?

3-Would this assay work on non-purified protein (the authors tested the assay with purified proteins)? Would it work on patient samples or environmental samples such as water? The assay should be tested in practice.

4- In the introduction section, the authors make the argument that that there is a need for the novel assay since ELISA has many steps; however, the HTRF assay has an incubation time of 18 hours. Hence, it takes time more time than RPLA or ELISA. Hence, what is the relevance of developing the novel assay when an ELISA yields results faster, is a tried-and-true assay and seemingly has a comparable range?

Validity of the findings

The authors did not provide statistical analyses and measures for the four pillars of a novel lab assay: accuracy, precision, specificity and sensitivity.

Additional comments

I recommend the authors evaluate their assay by comparing it other quantitative assays that are currently available. As a novel assay, it should be tested for its accuracy, precision, specificity and sensitivity as well with appropriate statistics being reported. On the other hand, I also recommend the authors clarify the relevance of developing the new assay particularly when "gold standards" of quantitative testing are available.

---

## Round 0.2 · Minor Revisions

Please address the Minor comments raised by Reviewer 2.

·

Basic reporting

The authors have generally addressed my comments pertaining to this section.

Experimental design

Although the authors have generally addressed my comments, I still believe that an established quantitative assay should be used for comparison purposes. The authors indicate that the ELISA is not available in Japan. Are there importing or financial restrictions that prevent the authors from obtaining this assay?

If there is a justifiable reason, then I suggest the authors at least acknowledge their awareness that accuracy, precision, specificity and sensitivity of their novel assay should be tested in comparison to an appropriate quantitative assay in their conclusions.

Validity of the findings

Same as the previous round of revision.

Additional comments

Nothing further to add.

---

## Round 0.3 · accepted · Accept

The authors properly addressed all the Reviewers' concerns.